# M2C: Towards Automatic Multimodal Manga Complement

**Hongcheng Guo*[1], Boyang Wang*[1], Jiaqi Bai[1], Jiaheng Liu[1],**
**Jian Yang[1], Zhoujun Li✉[1]**
[1]Beihang University
{hongchengguo,wangboyang,bjq,liujiaheng,jiaya,lizj}@buaa.edu.cn

## Abstract

Multimodal manga analysis focuses on enhancing manga understanding with visual and textual features, which has attracted considerable attention from both natural language processing and computer vision communities. Currently, most comics are hand-drawn and prone to problems such as missing pages, text contamination, and aging, resulting in missing comic text content and seriously hindering human comprehension. In other words, the Multimodal Manga Complement (**M2C**) task has not been investigated, which aims to handle the aforementioned issues by providing a shared semantic space for vision and language understanding. To this end, we first propose the Multimodal Manga Complement task by establishing a new M2C benchmark dataset covering two languages. First, we design a manga argumentation method called MCoT to mine event knowledge in comics with large language models. Then, an effective baseline FVP-M$^2$ using fine-grained visual prompts is proposed to support manga complement. Extensive experimental results show the effectiveness of FVP-M$^2$ method for Multimodal Mange Complement .

## 1 Introduction

Comics are enjoyed globally, often featuring a combination of text and illustrations. However, due to their hand-drawn nature, they are susceptible to damage during circulation, such as missing pages, text contamination, and aging. To address these issues, we propose a task called Multimodal Manga Complement (**M2C**), illustrated in Fig. 1. This task extends conventional text-based content complement by incorporating corresponding images as additional inputs to mitigate data sparsity and ambiguity (Ive et al., 2019a). Similar to other multimodal manga tasks (e.g., Multimodal Manga

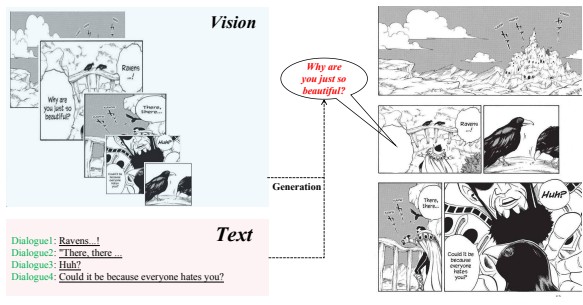

Figure 1: Motivation of our M2C task. The left part describes both visual and language modalities as the input (vision part depicts a sequence of images from a page of comics, and the language part is the corresponding text dialogues). The right part displays a page of comics and the text that we want to complete.

Translation (Hinami et al., 2021) and Comic Image Inpainting (Ono et al., 2021)), M2C aims to exploit the effectiveness of visual features for manga complement. Furthermore, M2C can serve as a foundation for other comic-related tasks.

However, as we can see in Fig. 1, compared to image caption (Vinyals et al., 2015) or other multimodal tasks, the comic data we use has its own characteristics. Here we conclude the differences and corresponding challenges: I). Comics have shorter conversational text lengths than other natural language forms, and contain many inflections or auxiliaries. This leads to a greater randomness in content generation that requires complement. II). The text and images of comics are complementary, often in pairs. This requires us to design better ways to exploit image features. III). Comic images and text contain timelines and implicit logic information, which shows us a challenge to leverage the logical knowledge to help improve the stability of generating complementary content. Therefore, in our work, we first propose the Multimodal Manga Complement (**M2C**) task to achieve the complement for with both vision and language information.

---

* First two authors contributed equally.
✉ Corresponding author.
https://github.com/HC-Guo/M2C

To eliminate the above limitations, inspired by recent CoTs (Wei et al., 2022; Wang et al., 2022), we first design the three-hop Manga Chain-of-Thought (MCoT) argumentation method to exploit timelines and logic knowledge in comics with large language models (Ouyang et al., 2022). It contains a three-hop inferring stage (Theme, opinion, and future). Then we propose a simple and effective **FVP-M**[2] method, including Feature encoding, Fine-grained visual prompt generation, and Dialogue complement. Specifically, in the token encoding stage, we use the pre-trained vision encoder to extract the visual tokens. Then, we follow (Johnson et al., 2017) to utilize the Transformer to encode the textual tokens. In FVPG, inspired by (Ying et al., 2021) and (Lin et al., 2020), we adopt a new way to aggregate global and local visual information. Specifically, the local visual features are aggregated in a self-attention way. while a mapping network in Fig. 4 is leveraged to generate the global visual feature, which will serve as a bias term added into the calculation of local visual features. Further, the outputs are the fine-grained visual prompts. After that, during the dialogue complement, following the works (e.g., ViLBERT (Lu et al., 2019)), we utilize co-Transformer to generate the vision-guided language tokens. Then the Transformer decoder is adopted to predict the results.

The contributions are summarized as follows:

- We first propose the Multimodal Manga Complement (M2C) task to handle the generation for the missing textual contents, which investigates the effect of the combination of vision and language modality.

- For M2C, we propose an effective Fine-grained visual prompt generation strategy for better aggregation of image information, and design a manga Chain-of-Thought method to exploit the logical information and the event knowledge implied by the comics.

- We establish a benchmark dataset on multimodal manga complement, and extensive experiments show the effect of our method.

## 2 Related Works

**Vision-Language Models**. The success of vision-language models can be attributed to the Transformer architecture (Vaswani et al., 2017; Radford et al., 2021; Li et al., 2020; Sharma et al., 2018; Miech et al., 2019; Bai et al., 2023b; Zhu et al.,

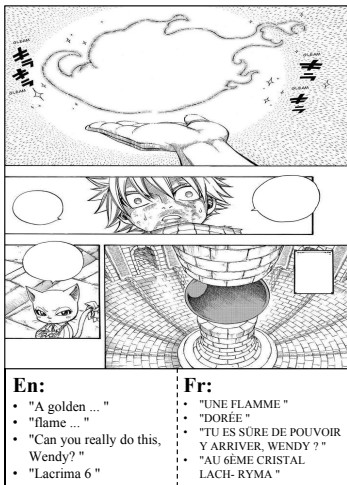

Figure 2: Example of an image with corresponding dialogues. En represents English and Fr represents French.

2023; Guo et al., 2022; Liu et al., 2022b,a) and large-scale training data (Liu et al., 2022c; Guo et al., 2023a,c; Bai et al., 2023a; Wang et al., 2023a; Guo et al., 2023b; Wang et al., 2023b; Yu et al., 2023). Transformer-based multimodal models incorporate both textual and visual information in a shared representation, which first preprocesses images using object detection models to extract information about specific regions or objects within the images. These region features are then projected into a common embedding space, which allows for seamless integration with the textual input. The multimodal Transformer architecture is then used to jointly process both the text and image inputs, using multi-head attention mechanisms to capture relationships and dependencies between the two modalities. This process results in the learning of a unified representation that captures the relationships between the textual and visual aspects of the data.

## 3 Datasets

We introduce the multimodal manga complement (M2C) benchmark dataset. Here, we described the details of the M2C dataset.

**Data Construction.** To ensure diversity in our comics, we have gathered open-source comic works in various styles and themes from across the Internet. Our dataset comprises 10,221 comic stories, each featuring several comic images and dialogues. To create the M2C corpus, we first employed the OCR model to extract sentences and then translated the English text into French using the multilingual translation model (Yang et al.,

## Three-hop Manga argument with CoT Prompting

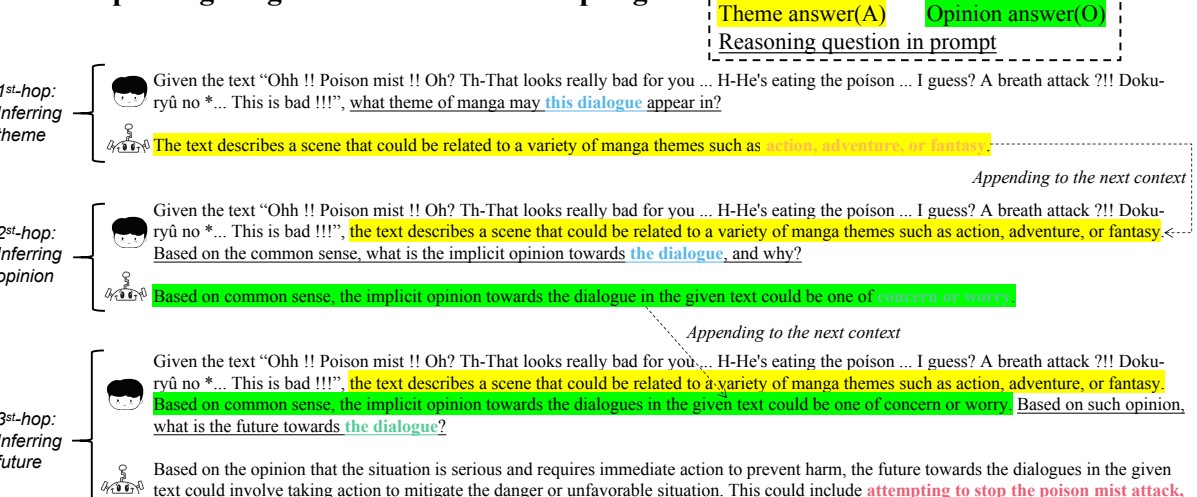

Figure 3: Manga Chain-of-Thought Augmentation.

2021a; Ma et al., 2020; Yang et al., 2021b). We then randomly selected one sentence from each story as the completion target while retaining the other dialogues as input, resulting in 45,995 pairs for our M2C benchmark. An example is provided in Fig. 2. The number of image-text pairs for training, validation, and testing data are 41995, 2000, 2000, respectively.

**Data Augmentation.** To exploit timelines and logic knowledge in comics and improve the generation performance, we design a three-hop manga Chain-of-Thought to facilitate step-by-step reasoning (inferring theme, opinion and future) as illustrated in Fig. 3.

## 4 Method

### 4.1 Multimodal Manga Complement

Supposing we have $K$ parallel sentences $\{(x_s^k, x_t^k)\}_{k=1}^K$ between source corpora $s$ and target corpora $t$, where $K$ is the number of training instances and each instance has the corresponding image $z_k$.

### 4.2 FVP-M$^2$

As shown in Fig. 4, our FVP-M$^2$ includes three stages: feature encoding, fine-grained visual prompt generation and dialogue complement.

### 4.2.1 Feature Encoding

For each image $z_k$, we directly use the vision backbone (e.g., vision branch of the widely-used CLIP model (Radford et al., 2021)) as the vision encoder

to extract the visual tokens for $z_k$ as follows:

$$\{v_m\}_{m=1}^M, v_c = \mathcal{H}(z_k), \qquad (1)$$

where $\mathcal{H}$ denotes the vision encoder and $M$ is the number of local cropped images in the whole picture. $v_c$ denotes the global visual feature encoded with the whole picture, while $v_m$ denotes the local feature encoded with a cropped image.

Similarly, given the source language $x_s^k$, based on the Transformer encoder $\mathcal{E}$, the source language tokens $\{s_f\}_{f=1}^F$ are extracted as follows:

$$\{s_f\}_{f=1}^F = \mathcal{E}(x_s^k), \qquad (2)$$

where $F$ is the number of source language tokens.

### 4.2.2 Fine-grained Visual Prompt Generation

In this stage, we design the fine-grained visual prompt generation method with the global visual feature $v_c$ and local feature $v_m$. Specifically, in Fig. 4, the $\{v_m\}_{m=1}^M$ is first aggregated by self-attention (Vaswani et al., 2017). Then we utilize a mapping network $\mathcal{M}$ implemented by two fully-connected layers with ReLU (Nair and Hinton, 2010) activation function to encode the global vision feature as follows:

$$\phi_{v_c} = \mathcal{M}(v_c). \qquad (3)$$

After that, we add $\phi_{v_c}$ as a bias term and generate the fine-grained visual prompt $p_v$:

$$p_v = \text{SF}(\frac{QK^T}{\sqrt{d/h}} + \phi_{v_c})V. \qquad (4)$$

where SF represents the softmax operation.

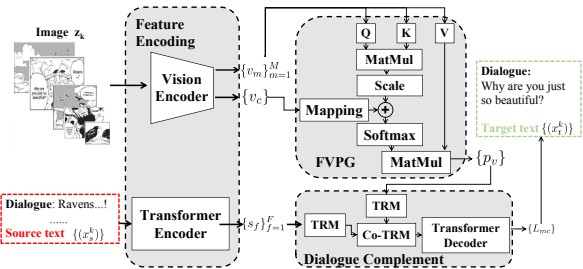

Figure 4: The overall framework of our proposed FVP-$M^2$ method for Multimodal Mange Complement task, which includes three stages (i.e., feature encoding, fine-grained visual prompt generation (FVPG), and dialogue complement). "TRM" and "Co-TRM" represent the Transformer and co-Transformer models, respectively.

### 4.2.3 Dialogue Complement

In Fig. 4, we utilize the Transformer module to fuse the information from other tokens within each modality for $\{s_f\}_{f=1}^F$ and $p_v$, respectively, and we represent the updated source language tokens and visual prompt as $\mathbf{S}$ and $\mathbf{P}$, respectively. Then, we take $\mathbf{S}$ as the query, and the $\mathbf{P}$ as the key and value in the co-attention module (Lu et al., 2019) to generate the vision-guided source language tokens $\{q_f\}_{f=1}^F$ as follows:

$$\{q_f\}_{f=1}^F = \mathop{\Big\|}_{h=1}^{H} \text{SF}\left( \frac{\phi_Q^h(\mathbf{S})\phi_K^h(\mathbf{P})^\top}{\sqrt{C}} \right) \phi_V^h(\mathbf{P}),$$
(5)

where $\|_{h=1}^H$ is the concatenation of the $H$ attentive features along the channel dimension. SF represents the softmax operation. $\phi_Q^h(\cdot), \phi_K^h(\cdot)$ and $\phi_V^h(\cdot)$ are the corresponding linear projection operations of the $h$-th head for the query, the key and the value, respectively. $C$ denotes feature dimension.

At inference, based on $\{q_f\}_{f=1}^F$, we use Transformer decoder to predict target sequence.

## 5 Experiments

### 5.1 Experimental Setting

**Implementation Details.** Our implementation is based on the Fairseq (Ott et al., 2019) toolbox. The model in Fig. 4 consists of 6 Transformer encoder/decoder layers. The number of attention heads is set as 12. We take the Adam optimizer (Kingma and Ba, 2015) with $\beta_1 = 0.9$ and $\beta_2 = 0.98$. The learning rate warms up from 1e-7 to 1e-4 in 2000 steps and then decays based on the inverse square root of the update number. The maximum number of tokens in each mini-batch is

| Model | En | Fr | Avg$_{all}$ |
|---|---|---|---|
| *Text-only Systems* | | | |
| Text Transformer (Vaswani et al., 2017) | 19.6 | 25.5 | 22.6 |
| ChatGPT (Ouyang et al., 2022) | 6.4 | 10.1 | 8.25 |
| ChatGPT (w/ MCoT) (Ouyang et al., 2022) | 15.2 | 18.8 | 17.2 |
| *Multimodal Systems* | | | |
| Deliberation Network (Ive et al., 2019b) | 21.4 | 26.7 | 24.1 |
| DCCN (Li et al., 2021) | 23.1 | 28.5 | 25.8 |
| Vision Matters (Li et al., 2021) | 22.5 | 27.4 | 24.9 |
| On Vision Features (Li et al., 2022) | 24.0 | 29.3 | 26.7 |
| FVP-$M^2$ (w/o FVPG & MCoT) | 22.9 | 28.4 | 25.7 |
| FVP-$M^2$ (w/o MCoT) | 27.2 | 30.9 | 29.0 |
| **FVP-$M^2$ (Our method)** | **29.9** | **33.7** | **31.8** |

Table 1: BLEU scores on M2C benchmark test set. Five baselines are compared by us. The bottom part shows the results of the models trained with text and vision modalities. The best results are highlighted.

1024. Dropout and label-smoothing rate are set as 0.3 and 0.1, respectively. We adopt the vision branch of CLIP based on the ViT-L/14 model. The effect of different vision backbones is shown in the appendix. All models are trained for 30 epochs and evaluated on one single linux machine with 4 NVIDIA A100 GPUs (80G).

**Data Quality Control.** We recruit 20 college students with a master's degree in English. To ensure the data quality, we conduct two rounds of data review. In the first review, these students carefully review all annotations to check whether each target sequence is consistent with the source sequence and the image. If not, they are asked to revise and provide the reason for revision with a confidence score. In the second round, samples without modification are considered correct. For these modified samples, the students are asked to discuss to determine the final version. For those controversial results, voting is adopted following the principle of majority rule.

**Evaluation.** We compute the cumulative 4-gram BLEU scores at inference. The beam search strategy is based on a beam size of 5 for the target sentence generation. The length penalty is 1.0.

**Baseline Methods.** As we are the first task in this area, we reproduce methods including Text Transformer (Vaswani et al., 2017), Deliberation Network (Ive et al., 2019b), DCCN (Lin et al., 2020), Vision matters (Li et al., 2021), and On Vision Features (Li et al., 2022). Besides, we also report the results of ChatGPT (Ouyang et al., 2022) and the variants of FVP-$M^2$.

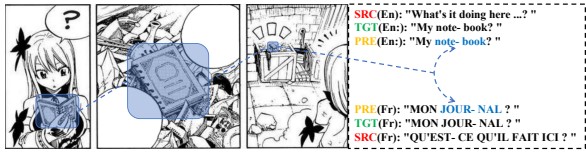

Figure 5: A qualitative example of two languages on English and French with the help of vision modality. Tokens in blue denote correct complement. SRC denotes the source corpus. PRE and TGT represent predicted results and ground-truth of target sentence, respectively.

## 5.2 Main Results

To demonstrate the effectiveness of FVP-M$^2$, we compare our method with baselines in Table 1. FVP-M$^2$ achieves the best BLEU scores in two languages. Specifically, FVP-M$^2$ outperforms by +9.2 BLEU scores on average compared with text-only Transformer, which demonstrates the effectiveness of vision for manga complement. Second, when faced with manga data containing little logical clues, ChatGPT performs poorly. Third, the effectiveness of our proposed FVPG and the MCoT method is verified through comparison.

## 5.3 Visulization

To further explore the necessity of visual modality for multimodal manga complement, we visualize the prediction results of a sample in Fig. 5. Specifically, the "note- book" (En) and "JOUR- NAL" (Fr) are masked in the target sentence, and these masked tokens describe the saliency regions in the corresponding left image. We have the following observations. We observe that even though the "note- book" is masked, the prediction results on this token are still right, which means that visual modality is complementary rather than redundant if the text is insufficient.

## 6 Conclusion

We first propose the Multimodal Manga Complement (M2C) task to handle issues such as missing pages, text contamination, and text aging in hand-drawn comics by providing a shared semantic space. We establish a new M2C benchmark dataset covering two languages. Then, we propose an effective FVP-M$^2$ method for the M2C task. Experimental results on M2C benchmark datasets show the effect of FVP-M$^2$ method.

## 7 Limitations

Although our proposed FVP-M$^2$ method has achieved substantial improvements for Multimodal Manga Complemnt, we find that there still exists some hyper-parameters (e.g., the number of encoder and decoder layers,) to tune for better results, which may be time-consuming. Besides, in our established datasets, we only support two languages currently, and we will extend to support more languages for Multilingual Multimodal Manga Complement in future work.

## 8 Acknowledgments

This work was supported in part by the National Natural Science Foundation of China (Grant Nos. 62276017, U1636211, 61672081), and the Fund of the State Key Laboratory of Software Development Environment (Grant No. SKLSDE-2021ZX-18).

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
