# OpenReview forum: "M2C: Towards Automatic Multimodal Manga Complement"
_EMNLP/2023/Conference — EMNLP 2023 Findings_

### Official Review · Reviewer_YEKH · 2023-08-03

**Typos Grammar Style And Presentation Improvements:** 1. Page 2, Line 082
**Soundness:** 3

**Excitement:**

4: Strong: This paper deepens the understanding of some phenomenon or lowers the barriers to an existing research direction.

**Missing References:**

BLIP-2 and CoCa and...

**Paper Topic And Main Contributions:**

The authors proposed a new task, dataset, and baseline called FVP-M2 for mulitmodal manga complement. More specific, local and gloal visual features are considered during feature encoding. Visual prompt generation is also used for better dialogue complement. Experiments on the proposed dataset show the effectness of the proposed method.

I found this work is interesting and useful. But I do have some key concerns on the loss function that consider En and Fr but during training only one language is considered. Besides, it is unclear how CoT is used.

The authors have introduced a novel task, dataset, and baseline referred to as FVP-M2, specifically for multimodal manga complementation. The method takes into account both local and global visual features during the feature encoding process and utilizes visual prompt generation for enhanced dialogue complementation. Experiments conducted on the newly proposed dataset demonstrate the effectiveness of this approach.

While I find the work intriguing and potentially valuable, I do have some crucial concerns that need to be addressed:

- The loss function mentioned in the paper takes into consideration both English (En) and French (Fr) language features. However, it's not clear from the paper how both languages are incorporated during training, especially since only one language appears to be considered. This discrepancy needs to be clarified.
- Additionally, the utilization of CoT (presumably a component or technique within the method) is not well explained. A more detailed description or rationale for its role within the model would add clarity and depth to the understanding of the proposed method.

**Questions For The Authors:**

- There appears to be an inconsistency in the data description. The authors have stated that they choose one sentence from each story and that the M2C dataset contains 10,221 stories. Therefore, logically, there would be 10,221 pairs, not 45,995. Could the authors please clarify this discrepancy and provide further details on how the pairs were constructed?

- Regarding Feature Encoding in Sec. 3.2.1, the description of "M is the number of local cropped images..." seems inaccurate. Based on the context, M should represent the local image features, such as grid features or visual patches, depending on the visual encoders (ResNet or ViT) being used. Consequently, M shouldn't be described as the number of cropped images. A clarification or correction of this point would be appreciated.

- The computation of p_v is not clearly explained in the paper. Providing details about how  p_v is computed, as well as information regarding its dimension, would be helpful for a full understanding of the method.

- Table 3 introduces some confusion regarding the difference between FVP-M2 map and FVPG. Since FVPG also appears to utilize the map function, it would be beneficial for readers if the authors could clarify the distinctions between these two aspects.

**Reasons To Accept:**

- The task of multimodal manga complementation is indeed engaging and valuable. The proposed dataset represents a promising resource that may prove useful for future research and developments within this domain.
- The model's pipeline, which incorporates visual features and visual prompts through the use of co-attention for sentence completion, is logically constructed. It appears to be a sensible approach to the problem at hand.
- The consideration of Manga Chain-of-Thought (CoT) within the method is a notable aspect, although I have some specific queries regarding this element, as detailed below.

**Reasons To Reject:**

- The loss function as expressed in Eq. (1) incorporates both source and target corpora. However, the pipeline appears to be operating based on only one corpus at a time, treating English (En) and French (Fr) separately. This inconsistency between the formulation of the loss function and the actual implementation needs to be addressed and clarified.
- The multimodal baselines employed in the paper do not represent the current state-of-the-art methods. Notable works such as CoCa, BLIP-2, and others are conspicuously absent from the comparison. Including these more recent and advanced methods would likely provide a more robust and meaningful evaluation of the proposed approach.
- Critical details are missing regarding the utilization of Manga Chain-of-Thought (CoT) within the FVP-M2 model. A comprehensive explanation of how CoT is integrated and its role within the overall approach would provide essential clarity and contribute to a more complete understanding of the model's functionality.

**Reproducibility:**

4: Could mostly reproduce the results, but there may be some variation because of sample variance or minor variations in their interpretation of the protocol or method.

**Reviewer Confidence:**

4: Quite sure. I tried to check the important points carefully. It's unlikely, though conceivable, that I missed something that should affect my ratings.

---

> ### Author Rebuttal · Authors · 2023-08-28
>
> ****Response to Review #3:****
>
> Thanks for your thoughtful and valuable comments.
>
> R3.Q1: **Data description.** Thanks for your suggestion. Here, to better utilize the corpora, we choose each sentence in a story as the target to predict. As the average length of each story is about 4.5, the number of total pairs is 45,995. We will revise this expression more clearly in our revised version.
>
> R3.Q2: **M in Feature Encoding in Sec. 3.2.1**. Thanks for your advice. We will revise M as the local image features in our new version.
>
> R3.Q3: **Computation of p_v.** The computation process of p_v follows the attention strategy with a bias item. Specifically, suppose M is the number of local image features. Then after the CLIP image encoder, we gain v_m (batchsize, M, Dimension=512) and v_c(bathsize, 1 , Dimension=512). So after the mapping network, the dimension of the $\phi$(Vc) is (batch, M, M). So the final dimension of p_v is (batchsize, M, Dimension), we will revise this explanation in our new version.
>
> R3.Q4: **FVP-M2 (map)**. In FVP-M2 (map), we directly generate visual prompts by encoding the visual tokens with simple linear layers. The name will be changed to avoid ambiguity in our revised version.
>
> R3.Q5: **Missing reference**. We will cite BLIP-2, CoCa and other related multimodal methods.
>
> R3.Q5: **Results of the SOTA multimodal methods.** We reproduce the results of the BLIP-2 and CoCa. Here, we extract features by using these two pre-trained models, and we observe that we can achieve better results when using better multimodal backbone methods, which further demonstrate the generalization ability and effectiveness of our method.
>
> | Model | En | Fr | Avg_{all} |
> | --- | --- | --- | --- |
> | CoCa [1] | 30.1 | 33.9 | 32.0 |
> | BLIP-2 [2] | 30.5 | 34.1 | 32.3 |
> | CLIP (our method) | 29.9 | 33.7 | 31.8 |
>
> R3.Q6: **More description about Manga Chain-of-Thought (MCoT)**. The text of comics generally does not contain a lot of logical information, which is not easy for complement according to the context. To exploit knowledge in comics and improve the generation performance, we design a data augmentation method for this task, which is a three-hop manga Chain-of-Thought to facilitate step-by-step reasoning (inferring theme, opinion and future). Specifically,  first, theme, opinion and future are the basic elements to create the comics. Second, for each hop, the generated text is appended to the next context as shown in Figure 3. We will revise the above description more clearly in the revised version.
>
> R3.Q7: **The loss function.** Thank you for your suggestion. Here, source corpora s and target corpora t are the same language, this kind of expression is merely to distinguish between input and output. To prevent ambiguity, we will specify that they belong to the same language in our new version.
>
> R3.Q8: **Typos Grammar Style And Presentation Improvements.** We will revise it in our new version.
>
> Reference:
>
> [1] CoCa: Contrastive Captioners are Image-Text Foundation Models
>
> [2] BLIP-2: Bootstrapping Language-Image Pre-training with Frozen Image Encoders and Large Language Models

---

### Official Review · Reviewer_4xZw · 2023-08-04

**Soundness:** 3

**Excitement:**

4: Strong: This paper deepens the understanding of some phenomenon or lowers the barriers to an existing research direction.

**Paper Topic And Main Contributions:**

Authors present a novel and new learning task named Multimodal Manga Complement (M2C) aimed at augmenting manga comprehension through the integration of visual and textual features. The primary objective of the M2C task is to address challenges arising from missing comic text content, which significantly hampers human understanding. Authors propose a shared semantic space for comprehending both visual and linguistic aspects. Additionally, a novel benchmark dataset for the future research in the multimodal manga complement is established in two languages. Overall, authors present a novel task serves as a foundational framework for addressing other comic-related tasks, marking a significant advancement in the field of manga analysis and comprehension.

**Questions For The Authors:**

1. Please provide substantive statistical data that justifies the significance of this problem, specifically by presenting statistical evidence concerning the prevalence of these issues within the comics field.

2. As numerous extant multi-modality studies propose the incorporation of a cross-modality attention layer to directly capture correlations between different modalities, I am wondering why the present work departs from employing such a conventional approach.

3. This work somehow shows a similar characteristics akin to a sequence-to-sequence language prediction task. In comparison to utilizing images as an additional modality, have you conducted an assessment of the viability of relying solely on the contextual relationships among dialogues for this particular task?

**Reasons To Accept:**

Overall, authors address a compelling and innovative problem of augmenting comics with missing dialogue or context. The study encompasses a fusion of linguistics and the correlation between language and image contextual semantics. By establishing a shared semantic space for vision and language understanding, the research explores the impact of combining vision and language modalities. This approach, known as the M2C task, holds significance in manga analysis as it enriches manga comprehension through visual and textual features, garnering considerable attention from natural language processing and computer vision communities.

Furthermore, the proposed model architecture exhibits promise in various other domains of application. The multi-modal complements model architecture could find utility in fields such as E-commerce or the streaming industry for complements recommendation. Future investigations might explore incorporating sequence products/videos titles or content summaries along with image signals. The model's potential could be harnessed to interpret the semantic interaction among products/videos and offer judgments for complements recommendation.

Lastly, authors' work stands is well-structured, and well justified by incorporation of appropriate baselines. The work's logic is easy to follow and understand. Additionally, they have contributed a new M2C benchmark dataset, encompassing two different languages, which will facilitate ongoing studies in the future.

**Reasons To Reject:**

Although, authors' proposed solution could address the issue of missing content in comics demonstrates novelty and commendable performance, the practicality of resolving this problem remains unsubstantiated. While my familiarity with this industry may be limited, it is essential for the authors to provide statistical evidence regarding the prevalence of such issues in the real world and how readers respond to such issues. This empirical approach would strengthen the application of the proposed solution and help establish its potential business value, after all this work is more likely to be application-oriented.

Considering the novelty of this work and the need for a more comprehensive evaluation, it is advisable to develop consolidated design criteria for evaluation metrics iteratively. Thus, in addition to the authors' proposed evaluation strategies, employing a human-in-the-loop evaluation approach would enhance the credibility of the work's robustness and reliability.

Lastly, there lacks sufficient ablation study to understand whether the image modality is truly needed for such work.

**Reproducibility:**

3: Could reproduce the results with some difficulty. The settings of parameters are underspecified or subjectively determined; the training/evaluation data are not widely available.

**Reviewer Confidence:**

4: Quite sure. I tried to check the important points carefully. It's unlikely, though conceivable, that I missed something that should affect my ratings.

---

> ### Author Rebuttal · Authors · 2023-08-28
>
> ****Response to Review #2:****
>
> Thanks for your thoughtful and valuable comments.
>
> R2.Q1: **Statistical data for the significance of the problem**. According to Wikipedia and industry research reports, the comic book industry has a very large base and geographically, Asia Pacific is the largest market, accounting for about 41% of the global market in 2019, with North America and Europe accounting for about 28% and 22%. By type, comic books mainly include physical comic books and digital comic books. Among them, the most widely used is physical comic books, accounting for approximately 89.90% of the total sales in 2019. In terms of application, comic books are mainly sold in retail stores, bookstores and online. And bookstores are the most widely used region, accounting for 49.01% of the total region in 2019. The global comic books market reached 25.1 billion dollars in 2020 and is expected to reach 31.5 billion dollars in 2026, at a compound annual growth rate (CAGR) of 3.3%. In such a huge market, the damage to comics is very serious. Due to aging and vandalism, the damage rate of comics is as high as 72.4%. In 2020, for example, the loss due to damage is as high as around 14.6 billion dollars. At the same time, the lack of comics will also indirectly breed piracy and infringement of copyright, and the annual cost of comics infringement and maintenance of action is up to tens of billions of dollars. To sum up, the task of comic complement is of great significance to the protection of intellectual property rights and the reduction of losses in the industry.
>
> R2.Q2: **Cross-modality attention layer**. We have already adopted the co-attention module in our framework. In Line 92-93 and Figure 4 in our paper, the Co-Transformer module represents the cross-modality attention layer, which has been proposed in ViLBERT,a model for learning task-agnostic joint representations of image content and natural language. We will present more clearly in new version.
>
> R2.Q3: **Rely solely on the contextual relationships**. Please refer to the Text Transformer in Table 1 in our main paper. We have already given the results with only the text components (i.e., the results of Text Transformer). We will mention these results more clearly in our new version.
>
> | Model | En | Fr | Avg_{all} |
> | --- | --- | --- | --- |
> | Text Transformer | 19.6 | 25.5 | 22.6 |
>
> R2.Q4: **Human-in-the-loop evaluation**. We design questionnaires for human assessment. First, we recruited 10 college students from different departmemts for a fair and diverse evaluation with comic book reading experience as raters, and then we randomly selected 15 examples from the test set as a questionnaire. For each test case, at least three labers randomly are selected to give the independent scores and then we summarize the scoring results for a objective evaluation. The college students are required to score different models. The scores are from 1 to 5 (1: ”very bad”, 2: “bad”, 3:“borderline”, 4: “good”, 5:“very good”). There are two criteria for scoring: linguistic coherence and fun.  Finally, we calculate the average score and rank the models as follows, and we observe that our method ranks first, ChatGPT is not the lowest ranked while it has the lowest BLEU score, it has a high score in language coherence, but lacks the fun of comics.
>
> The average scores of 10 college students for baseline methods:
>
> |  | ChatGPT | DCCN | On Vision Features | FVP-M$^{2}$ |
> | --- | --- | --- | --- | --- |
> | Language coherence | 4.37 | 4.06 | 4.13 | 4.39 |
> | Fun | 3.71 | 3.85 | 4.01 | 4.32 |
>
> The ranking of four methods:
>
> | Model | Ranking (Language coherence) | Ranking (Fun) |
> | --- | --- | --- |
> | ChatGPT | 2 | 4 |
> | DCCN | 4 | 3 |
> | On Vision Features | 3 | 2 |
> | Our method | 1 | 1 |

---

### Official Review · Reviewer_naWw · 2023-08-13

**Typos Grammar Style And Presentation Improvements:** 1. Line 86, W for while should be cap…
**Soundness:** 4

**Excitement:**

3: Ambivalent: It has merits (e.g., it reports state-of-the-art results, the idea is nice), but there are key weaknesses (e.g., it describes incremental work), and it can significantly benefit from another round of revision. However, I won't object to accepting it if my co-reviewers champion it.

**Paper Topic And Main Contributions:**

The paper proposes the task of Multimodal Manga Complement (M2C) which is the automatic understanding of manga (comics) with visual and textual features, so as to complete the missing text when the images are present.

This paper introduces a new dataset (M2C benchmark dataset) consisting of two languages, English and French. The text is extracted using the OCR model (TrOCR). The french component is achieved by using a XLM-R model (En-Fr).

This paper introduces a novel Manga argumentation method called MCoT to mine event knowledge in comics using large language models.

Establishes a baseline (FVP-M^2) for the task.

**Reasons To Accept:**

The methodology is clearly defined.

The paper presents the task challenge well, namely e.g; comics have shorter conversational text lengths, which makes it harder to make predictions.

The methodology presented exploits the complementary textual and visual information present in the comics. For this the authors propose the FVP-M^2 method which includes three components: (1) feature encoding, (2) fine-grained visual prompt generation, and (3) dialogue complement.

Secondly the logical information present in the timelines and visual/text is exploited. This is done using Three-hop Manga Chain-of-Thought (MCot) which is a three hop inferring stage (theme, opinion and future).This is explained well in Figure 3.

**Reasons To Reject:**

The fine-grained visual prompt generation is not clear. Ablation study on the prompt generation is needed to understand the importance of this component. Table 2 presents results for models w/o FVPG & MCoT and model w/o MCot. However model w/o FVPG is not explored, i.e dialogue component directly using the transformer encoder and the vision encoder outputs, without the FVPG. [This result is presented in the rebuttal]

What happens if only the text components are used for the dialogue prediction? I.e no multimodality.

Could other methods like image captioning be applied here?

**Reproducibility:**

3: Could reproduce the results with some difficulty. The settings of parameters are underspecified or subjectively determined; the training/evaluation data are not widely available.

**Reviewer Confidence:**

4: Quite sure. I tried to check the important points carefully. It's unlikely, though conceivable, that I missed something that should affect my ratings.

---

> ### Author Rebuttal · Authors · 2023-08-28
>
> ****Response to Review #1:****
>
> Thanks for your thoughtful and valuable comments.
>
> R1.Q1: **Results w/o FVPG**. For the model without FVPG, we directly use the transformer encoder and the vision encoder outputs as follows, and we observe that FVP-M$^{2}$ (w/o FVPG) are lower than FVP-M$^{2}$, which shows the effectiveness of our FVPG. In our new version, we will add this result to better explain the effectiveness of our method.
>
> | Model | En | Fr | Avg_{all} |
> | --- | --- | --- | --- |
> | FVP-M$^{2}$ (w/o FVPG $\&$ MCoT) | 22.9 | 28.4 | 25.7 |
> | FVP-M$^{2}$ (w/o MCoT) | 27.2 | 30.9 | 29.0 |
> | FVP-M$^{2}$ (w/o FVPG) | 26.6 | 29.1 | 27.9 |
> | FVP-M$^{2}$ (Our method) | 29.9 | 33.7 | 31.8 |
>
> R1.Q2: **Results with only the text components.** Please refer to the Text Transformer in Table 1. We have already given the results with only the text components. We will mention the following results more clearly in our new version.
>
> | Model | En | Fr | Avg_{all} |
> | --- | --- | --- | --- |
> | Text Transformer | 19.6 | 25.5 | 22.6 |
>
> R1.Q3: **Results with image caption methods.** We replicate the current SOTA approaches for image caption (SimVLM[1], OFA[2]) as follows, and we observe that the performance results of these methods are lower than a lot. We assume that these caption methods focus more on attributes (e.g., objects in the image) and scenes, while the style of the manga is more about feelings and plot-related. Therefore, the domain of manga is not compatible with these image caption methods. In our new version, we will revise the above discussion and try to extend our method on the more backbone models to facilitate the advancement of the community, such as OFA and SimVLM.
>
> | Model | En | Fr | Avg_{all} |
> | --- | --- | --- | --- |
> | SimVLM [1] | 5.6 | 8.4 | 7.0 |
> | OFA [2] | 6.0 | 9.3 | 7.7 |
> | FVP-M$^{2}$ (Our method) | 29.9 | 33.7 | 31.8 |
>
> R1.Q4: **Typos Grammar Style And Presentation Improvements.** We will revise these typos and improve the presentation in our new version.
>
> Reference:
>
> [1] Simvlm: Simple visual language model pretraining with weak supervision.
>
> [2] OFA: unifying architectures, tasks, and modalities through a simple sequence-tosequence learning framework.

---

### Meta-Review · Area_Chair_HyHT · 2023-09-19

**Recommendation:** 4

**Metareview:**

This paper investigates manga understanding using visual and textual features (completing the missing text when only the visual cues are available). The research is original and the methodology is clear.

Pros:
* The paper shows how to use the complementary textual and visual information present in the comics
* The methodology that includes visual features and visual prompt using co-attention for sentence completion

Cons:
* Lack of statistical evidence that supports the claims in a real scenario
* Lack of ablation studies

---

### Decision · Program_Chairs · 2023-10-07

**Decision:**

Accept-Findings

**Comment:**

This paper investigates manga understanding using visual and textual features (completing the missing text when only the visual cues are available). The research is original and the methodology is clear.

Pros:
* The paper shows how to use the complementary textual and visual information present in the comics
* The methodology that includes visual features and visual prompt using co-attention for sentence completion

Cons:
* Lack of statistical evidence that supports the claims in a real scenario
* Lack of ablation studies